# How does a pair of near-vision spectacle correction empower older Zanzibari craftswomen?: A qualitative study on perception

Michelle Fernandes Martins[1‡], Fatma Omar[2‡], Omar Othman[2‡], Gianni Virgili[1], Ai Chee Yong[1], Damaris Mulewa[3], Christine Graham[1], Carlos Price-Sanchez[1], Ronnie Graham[4], Adrianna Farmer[1], Eden Mashayo[5‡], Ving Fai Chan[1,4,6‡]*

1 Centre for Public Health, School of Medicine, Dentistry and Biomedical Sciences, Queen's University Belfast, Northern Ireland, United Kingdom, 2 Ministry of Health, Zanzibar, Tanzania, 3 Independent Researcher, Nairobi, Kenya, 4 Vision Action, London, United Kingdom, 5 Vision Care Foundation, Dar-es-Salaam, Tanzania, 6 College of Health Sciences, University of KwaZulu-Natal, Durban, South Africa

‡ MFM, FO and OO are the co-first authors who have contributed equally to the research and manuscript writing. VFC and EM are joint senior authors on this work.
* v.chan@qub.ac.uk

⊘ OPEN ACCESS

**Data Availability Statement:** All dataset files are available from the Queen's University Belfast PURE Repository database (https://doi.org/10.17034/d32d197f-87fa-4c4c-bc08-b80e2e1c366d).

## Abstract

### Background

Studies have shown that correcting presbyopia among women could increase short-term income and quality of life. However, it is unclear whether these short-term outcomes translate to long-term empowerment. This is partly due to women's empowerment being understudied in the eye health field. Hence, we attempted to understand Zanzibari craftswomen's perception of how near-vision spectacle correction could empower them.

### Methods

Semi-structured interviews were conducted with 24 craftswomen with presbyopia (7 to 21 April 2022), identified from Zanzibari cooperatives using quota and heterogeneity sampling. We included a sample of tailors, beaders/weavers, and potters who were 40 years and older. Directed content analysis was performed on interview transcripts.

### Results

Two themes and seven sub-themes emerged from the data. Craftswomen perceived that at the personal level, near-vision spectacle correction could improve their economic empowerment (better income and savings and buying things for themselves), psychological empowerment (more self-confidence and decision-making), political empowerment (taking up leadership roles), and educational empowerment (acquiring new skills). At a relational level, they perceived that near-vision spectacle correction could bring about economic empowerment (ability to buy things for the family), social empowerment (ability to participate in social activities), and educational empowerment (ability to educate other women).

**Funding:** YES. VFC. NPO 6240 R8898CPH Novartis (Excellence in Ophthalmology and Vision Award, XOVA) https://www.xova.novartis.com/#dwell The funder did not play any role in the study design, data collection and analysis, decision to publish, or preparation of the manuscript.

**Competing interests:** This study formed part of MFM's master's dissertation. This does not alter our adherence to PLOS ONE policies on sharing data and materials.

## Conclusion

Older craftswomen perceived that correcting near vision could empower them at personal and relational levels that encompass economic, psychological, social, political and educational empowerment. The findings laid the foundation for future research into eye health and women's empowerment.

## Introduction

Presbyopia is one of the most common causes of vision impairment, affecting 1.8 billion people despite being easily and inexpensively treated with a pair of spectacles [1]. Presbyopia is an age-related near vision impairment that commonly develops around 40 years of age—when individuals are still within the workforce. Thus, uncorrected presbyopia has great economic implications [2]. The estimated global productivity losses amongst working-age presbyopic adults amount to US\$ 25 billion annually [3]. The rates of uncorrected presbyopia in Sub-Saharan Africa could be more than 80%, [4] with an overall effective spectacle coverage of only 5.7% in 2022 [5]. Additionally, uncorrected presbyopia disproportionately affects 11% more women than men [4, 6]. Studies have attributed this disparity to women having higher life expectancy [4] than men, as well as facing greater financial, social, and cultural barriers to eye health services [7]. Despite presbyopia's disproportionate economic and health impacts among women in low-resource settings, slow progress is seen in women-targeted refractive error programmes [6].

Studies on the impact of presbyopia and its correction on women have to date focused on outcomes such as work productivity, [8, 9] income, [10] visual function [11] and quality of life (QoL) [9, 11–13]. For example, a cohort study of female textile workers in Bangladesh showed that uncorrected presbyopia is associated with earning \$6.51 less per month than those with correction and no vision loss [10]. Studies among textile workers in South Africa [13] and tea pickers in India [8] found improvement in work productivity of 6.4% and 22%, respectively, after correcting presbyopia. A study in Zanzibar also demonstrated that correcting presbyopia improved the visual function of near activities, such as reading small print and threading a needle [11]. The same study [11] and others [9, 12] have also reported improvement in the QoL scores post-correction. However, it is unclear whether the improvements in short-term outcomes could lead to empowering women in the longer term.

Empowerment is "the process of enhancing an individual's or group's capacity to make purposive choices and to transform those choices into desired actions and outcomes" [14]. Women's empowerment seeks to provide women opportunities equal to those of men and increase their self-reliance. In practical terms, this refers to improving financial and economic status in society and ensuring their ability to participate in decision-making at all levels (household, local, and government). Women's empowerment can therefore be seen as a shift from a lack of power amongst women and girls to that of more power [15]. Pratley's systematic review [16] found 67 eligible studies that suggest maternal and child health outcomes positively affect women's empowerment. The authors concluded that with 121 outcome indicators for women's empowerment identified, it is difficult to measure the impact of women's empowerment due to the "lack of a clear definition of the concept and direct indicators of all dimensions of women's empowerment" [16–18] at the personal, relational and environmental levels.

Of the 1.62 million population in Zanzibar, 51.4% are women [19]. Further, 28.5% of these women are of working age, and nearly a quarter of Zanzibari women (23.7%) head their

households [19]. Evidence from Tanzania [20] and Zanzibar [21] suggests that cultural exclusion and safety issues have led to women's poor participation in education, employment, and politics. In response, cooperatives have become a platform for older women to make a living, mainly as craftswomen. About 9.3% of women who head households in urban areas and 3.4% in rural areas are craftswomen [19]. Craftwork such as beading, weaving, pottery, and soap making requires good near vision. Hence, older craftswomen (40 years and older) may suffer reduced productivity due to presbyopia and vision impairment.

A populational-based study found a crude prevalence of presbyopia as 89.2% amongst those over 40, with a low correction rate of 17.6% [11]. The two most common barriers reported were that individuals did not prioritise correction and could not afford to pay for spectacles [11]. Another study from mainland Tanzania [22] further demonstrated that near-vision spectacle improved visual function and QoL. Nevertheless, no study has explored whether these corrections have empowered beneficiaries. Hence, this study aims to understand Zanzibari craftswomen's perceptions of presbyopia correction and how it could empower them. These findings could help develop indicators for measuring women's empowerment in this setting.

## Methods and material

This was a qualitative research which employed semi-structured interviews.

### Sample and sampling

A combination of quota and heterogeneity sampling were used to identify potentially significant variations among the participants [23]. To investigate the impact of presbyopia correction on older craftswomen's self-defined empowerment, we only included women from the sample population (n = 228) who had a) correctable distance and near vision impairment with spectacles and b) correctable near vision impairment with spectacles with no distance vision impairment. Interviews were held on the day of vision screening before spectacles were provided to reduce the likelihood of the intervention altering their perceptions.

We decided on a sample size of 24 participants, even though 16 would likely reach 90% of data saturation [24, 25]. The larger sample size increased the representation of women living on Unguja and Pemba islands (Pemba having a higher poverty rate than Unguja), [26] of different age groups, and engaged in different types of work (degrees of near vision demand of a tailor > beader/weaver > potter). This allowed for greater transferability of this study to other contexts [26]. Table 1 shows the sampling frame used to guide the intentional selection of the 27 craftswomen across 10 of 19 cooperatives from the WE-ZACE study. Three craftswomen declined to participate in the interviews as they were busy with other activities.

**Table 1. Sampling frame for participants and the actual number of craftswomen who participated in the semi-structured interviews.**

| Location | Unguja island | | | | | | Pemba island | | | | | |
|---|---|---|---|---|---|---|---|---|---|---|---|---|
| Age (years) | 40–55 | | | >55 | | | 40–55 | | | >55 | | |
| The main type of crafts | Beading/ weaving | Pottery | Tailoring | Beading/ weaving | Pottery | Tailoring | Beading/ weaving | Pottery | Tailoring | Beading/ weaving | Pottery | Tailoring |
| Proposed number of craftswomen | 2 | 2 | 2 | 2 | 2 | 2 | 2 | 2 | 2 | 2 | 2 | 2 |
| The actual number of craftswomen | 3 | 3 | 3 | 2 | 1 | 0 | 3 | 3 | 4 | 0 | 1 | 1 |

**Table 2. Definitions of women's empowerment by VeneKlasen and Miller [15], Lombardini, Bowman and Garwood [27] and Mandal [28].**

| Author, year | Definitions |
|---|---|
| VeneKlasen and Miller [15], 2002 | • 'Power over': Often seen from a negative lens where someone or a group of people dominate another person or group of individuals.<br>• 'Power within' (intrinsic agency): An individual's awareness of their worth and knowledge.<br>• 'Power to' (instrumental agency): A person's ability to mould their own life.<br>• 'Power with' (collective agency): Respectfully collaborating with others to reach a common goal. 'Power with' encourages improvements in social relationships. |
| Lombardini, Bowman and Garwood [27], 2017 | • Personal: Aspects relating to a woman's self-image, decision-making capabilities and acting on these decisions, perceptions of the part she plays in society and other women's roles.<br>• Relational: Looks at the woman's relationships around her, which involves relationships within her household, community members, local markets, community/council leaders and those with the capacity to make decisions within a community.<br>• Environmental: Looks at a wide-ranging context within the broader society that deals with either formal or informal changes. For example, legal and political changes (formal) or changes in societal values and opinions. |
| Mandal [28], 2013 | • Economic empowerment: Underprivileged individuals in all societies are provided with opportunities to be liberated from oppression and deprivation, which results in economic self-sufficiency. These provisions allow individuals and households to enjoy a free market and afford basic needs such as nutritious meals, housing, clean water and medicines.<br>• Educational empowerment: Getting an education in one or more forms. It can be related to learning new information or skills in various areas. For example, basic education such as reading and writing; other forms of schooling; education related to employment; work-related skills; politics; the legal system; or any other form of learning.<br>• Political empowerment: Participating in the political process as an individual or a group at a community, regional or national level. Participating in the political process can entail any of the following: The ability/freedom to vote; taking part in political/leadership campaigns; taking part in activities within a community that deals with making decisions that can affect the community or society that they serve; and lastly, the ability to communicate with government officials about specific social problems.<br>• Psychological empowerment: Making decisions in different areas of her life, such as how to use her income, health-related decisions, decisions within her household, etc. Psychological empowerment may also involve self-worth recognition, the improvement of self-confidence, and attitudes about violence against women.<br>• Social empowerment: Enhancing women's status in society and their relationships with others. Social empowerment is also about women enjoying the same rights as men and having equal standing in society. |

### Research team

Semi-structured interviews were conducted by two Swahili-speaking field researchers from Zanzibar (FO, female, national eyecare coordinator, 18 years experience in qualitative research; and OO, male, data analyst, 6 years experience in qualitative research) who understood the local culture well enough to facilitate meaningful conversations with participants. Both interviewers had previously discussed the aims and objectives of the WE-ZACE project with cooperative managers.

### Data collection

The interview guide was designed by the research team in English and then translated into Swahili by an anthropologist (CG), which was then back-translated into English by a public health optometrist (EM) to ensure correct translation. The guide was piloted with 20 craftswomen with minimal adjustments made. The main, open-ended question asked was, "How do you think correcting your near vision could help empower you (change your life positively)?" followed by further probing questions.

The semi-structured interviews were conducted from 7 to 21 April 2022, and audio-recorded by FO and OO in a private room, away from other people, with no third party present. The interviews took 12–24 minutes per participant. The interviews were then manually transcribed verbatim and translated into English by the interviewers. Member checking of

transcripts was done using a debriefing session with craftswomen after the interview, where they checked their transcripts, thereby improving the confirmability of the research findings. Transcripts were then translated back into Swahili by another translator (DM) to ensure the information's accuracy and that the data's richness was retained.

## Data analysis

Several authors have tried to understand this concept. VeneKlasen and Miller [15], for example, explored this by looking at the expression of power; Oxfam's Women's Empowerment Index framework by Lombardini [27] recognises that changes in women's empowerment can occur at an individual, relational or environmental level; Mandal [28] further categorised women's empowerment into five categories: social, psychological, educational, economic and political. These definitions are shown in Table 2.

Our research used a deductive and semantic approach to conduct direct content analysis to create themes, categories and sub-categories based on VeneKlasen and Miller [15] and Mandal's [28] definition of women's empowerment. Relevant information from the English translation of the transcripts was manually extracted onto Microsoft Excel® Spreadsheets. Two analysts (MFM, a medical doctor and Master's student; DM, a Swahili-speaking global health practitioner with seven years of qualitative research experience) independently coded the transcripts. The data analysts made notes during the coding process, which assisted with reflexivity. Additionally, making notes during coding aided the data analysts during consensus discussions. The data analysts used an iterative approach to reach a consensus on coding by reading and re-reading the data from the transcripts to familiarise themselves with the data. The data analysts independently analysed the first five transcripts to develop codes and then met to resolve any disagreements on data coding. The remainder of the transcripts were independently coded by both data analysts, who used the agreed-upon themes, subthemes and sub-categories from the first five transcripts as references. However, new themes, subthemes and sub-categories emerged at this coding stage. After completing the coding of all 24 transcripts, both analysts discussed disagreements to reach a consensus. A third data analyst (VFC) assisted with the refinement of themes and subthemes after each stage of consensus between MFM and DM. The subthemes were accompanied by the participants' quotes, which their profiles described and abbreviated to their vision task_age_island of residence (e.g. Weaving_45_Ung).

This study received approvals from the Ethics Committees from the Zanzibar Human Research Institute (ZAHREC/04/PR/MARCH/2022/12), Zanzibar Office of Government Chief Statistician (6221C2601263D) and Queen's University Belfast (MHLS 22_72). We obtained the participants' informed consent before the interviews were conducted. The consolidated criteria for reporting qualitative studies (COREQ) checklist [29] was primarily used to ensure appropriate data reporting. Supporting information on inclusivity in global research in included in S1 File.

## Results

The age of craftswomen interviewed ranged from 40 to 63 years, with a mean age of 49. There were equal numbers of weavers/beaders, potters and tailors in the sample (n = 8 each), with 12 from each island. Data analysis revealed two themes and seven subthemes (Fig 1).

### Theme 1: At a personal level, near-vision spectacle correction could lead to economic, psychological, political and educational empowerment

**Subtheme 1: Near-vision spectacle correction could bring about economic benefits.** The craftswomen explained that near-vision spectacle correction could improve their work

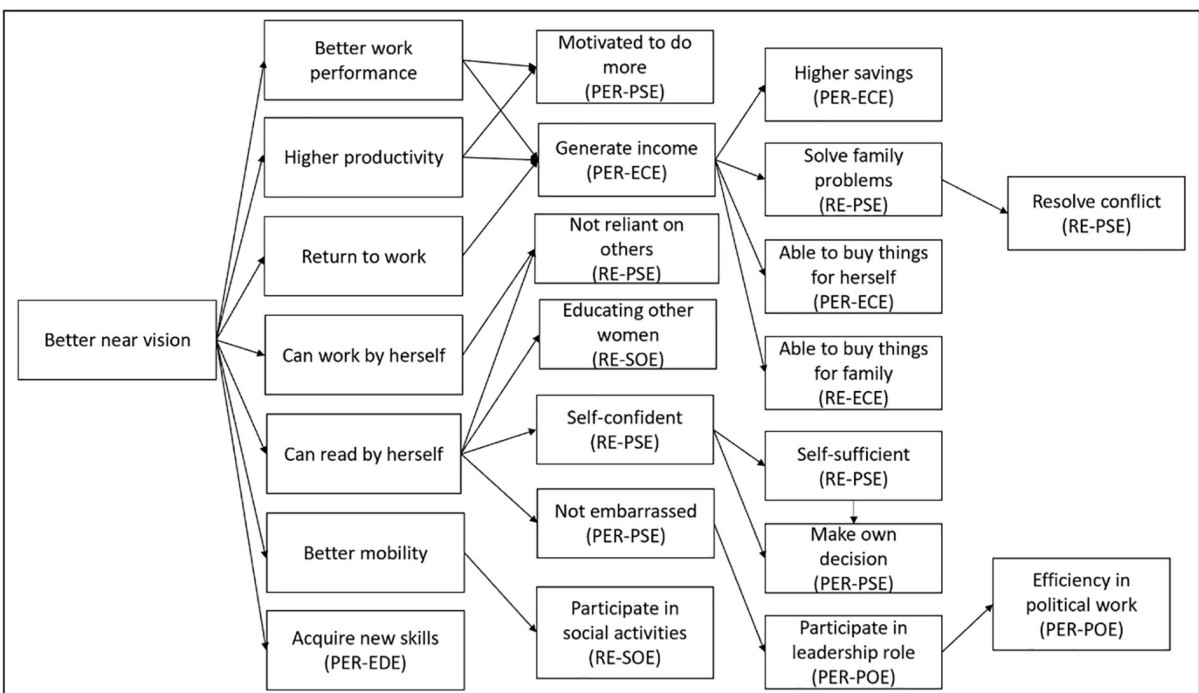

**Fig 1. Visual map of the perceived impact of near-vision eyeglasses correction on women's empowerment.** Key: PER-EDE = Personal level Educational Empowerment; PER-PSE = Personal level Psychological Empowerment; PER-ECE = Personal level Economic Empowerment; PER-POE = Personal level Political Empowerment; RE-PSE = Relational level Psychological Empowerment; RE-SOE = Relational level Social Empowerment; RE-PSE = Relational level Psychological Empowerment; RE-SOE = Relational level Social Empowerment; RE-ECE = Relational level Economic Empowerment.

productivity and increase their craft-based income. [*Weaving_49_Ung*: *"these (spectacles) will help to improve my eye health as well as increase production efficiency and [also] increase revenue."*]. Some craftswomen who had to stop working due to poor near vision could return to work after their vision was corrected [*Weaving_53_Pem*: *"When I return to the group, I will work harder and sell more. I will be able to be more active at home, outside of group time, by selling some of my work and generating an income."*]. Improvement in income may also lead to higher savings and the ability to buy things they wish to, which they were unable/reluctant to do before [*Tailoring_44_Ung*: *"I will be able to keep more savings as I will be able to work more."; Tailoring_40_Pem*: *"buy things for me."*].

**Subtheme 2: Near-vision spectacle correction could bring about psychological improvement.** Craftswomen perceived that with improved near vision, they would be more self-sufficient and independent [Weaving_40_Ung: *"To be empowered in terms of livelihood, e.g. right now I have been given a pair of reading glasses. Now I can read my Zanzibar ID numbers. Now I can read by myself along with doing other crafts at our cooperative."*]. They would also be more confident, giving them the autonomy to make decisions for themselves [*Weaving_52_Pem*: *"Now I will be more confident and be able to make my own decision."; Tailoring_46_Pem*: *"I was somehow in a state of worry on making a lot of decision[s] because of my eyesight (answering a question about changes in the ability to decide your affairs after getting spectacle)."*]. Some mentioned that after near-vision spectacle correction, they would no longer feel embarrassed or ashamed because they of their inability to recite the Qu'ran correctly [*Tailoring_40_Pem*: *"made mistakes or seemed illiterate"*].

Furthermore, near-vision spectacle correction increased motivation to work [*Tailoring_44_Ung*: *". . . I will be able to do my work effectively and properly and I felt motivated to work more."*], more job satisfaction [*Tailoring_48_Pem*: *". . . getting eyeglasses that will really [make] us happy in our working."*] and accomplish new skills [*Weaving_41_Pem*: *"I can now do knitting and crocheting and SMS reading."*]

**Subtheme 3: Near-vision spectacle correction could lead to leadership participation.** Many craftswomen believed their near-vision spectacle correction would encourage them to compete for leadership roles now that they no longer had reading difficulties which caused them embarrassment [*Weaving_41_Pem*: *"Now, I think I will be frontline in competing for the position since there will be no chance of being [embarrassed] because of not being able to read."*; *Weaving_40_Ung*: *"Yes, before I was involved. . .[in] leadership within the community and also Chairperson in leading women, and now I want to be involved in various positions [within] women's leadership."*]. Those already in leadership roles believed that correcting presbyopia would advantage them by increasing the effectiveness of their current political work [*Weaving_58_Ung*: *"Yes, I will be able to read and do [my political work more] effectively."*; *Tailoring_57_Pem*: *"Now, I am a leader in my corporative, so it is ok that [my vision correction] will give me an added advantage."*].

**Subtheme 4: Near-vision spectacle correction provided an opportunity to learn something new.** Craftswomen's participation in this study allowed them to learn more about eye health, where they learnt that one should have good vision for both near and distance vision. [*Tailoring_48_Pem*: *". . . I would like to say today I am very grateful for being educated just knowing that the vision capacity can also be known for the ability to read distant objects. . ."*].

## Theme 2: At a relational level, near-vision spectacle correction will improve social empowerment in the community and at home

**Subtheme 1: Near-vision spectacle correction could lead to stronger social connection and reduce relational conflict.** Many craftswomen avoid participating in social activities due to their poor near vision. Presbyopia correction would allow craftswomen to engage more in community gatherings [*Pottery_45_Pem*: *"Yes, my involvement in social activities will increase because I was avoiding some of the journeys because of my eyesight problem."*]. Furthermore, improved near vision would decrease reliance on family members and reduce conflicts associated with frequently asking for help [*Tailoring_48_Pem*: *"From this moment, there is no need to quarrel with the children to help thread the sewing needle."*].

**Subtheme 2: Near-vision spectacle correction would allow craftswomen to financially provide more for their families.** Craftswomen believed that near-vision correction would allow them to buy things for their families and to contribute more money to the household [*Pottery_45_Ung*: *"I will be able to buy things for my kids and the family too and be able to solve some problems at home."*].

**Subtheme 3: Near-vision spectacle correction would enable craftswomen to educate other women.** Craftswomen were enthusiastic about motivating and educating other women. One craftswoman expressed how she would pass on her knowledge to other women about saving money [*Tailoring_44_Ung*: *"Yes, as I will be able to read and give other [women] education so as they can also be motivated and get well informed."*; *Weaving_58_Ung*: *"for instance, in case of a medical emergency, I can use my savings. . .[with] regards [to this], I will educate more women to do so."*].

## Discussion

This study aimed to explore what craftswomen perceived the impact near-vision spectacle correction could have on women's empowerment. The findings show that better near vision has far-reaching perceived benefits as it allows the craftswomen to have better autonomy in their lives and opens up a greater array of life choices at a personal and/or relational level. These findings have great implications as women's empowerment is vital to achieving gender equality and poverty reduction (2030 Agenda for Sustainable Development).

Economic empowerment via improved income and savings remained the predominant theme, as most quotes were related to personal economic benefits. This is unsurprising as economic empowerment could lead to better access to markets, resources, banking services, asset acquisition, and employment opportunities. Currently, Zanzibari women face challenges in improving their basic material conditions, such as obtaining a bank account and owning assets. For example, only 25% of women in Zanzibar own their land [19]. Craftswomen's perceptions of increased income and improved productivity are supported by quantitative data analyses from eye health studies in LMICs [9, 11, 13, 30].

Eckert et al.'s economic modelling estimated a 30% productivity loss for people with moderate to severe vision impairment, resulting in 30%–55% reduced productivity among the unemployed 50–64 age group [31]. The literature also suggests that older adults who lose their vision face greater disadvantages than those with lifelong vision loss, [31–33] and that vision loss can contribute to early departure from the workforce [34, 35]. This might be the case among presbyopic craftswomen who stopped working as their near vision was too poor to cope with the precision of their crafts. Hence, with near-vision spectacle correction, Zanzibari craftswomen could re-participate in their craftwork and significantly improve their lives.

The ability to generate income would increase craftswomen's economic freedom and subsequently improve their well-being [36]. Economic freedom benefits both women and their families. At a personal level, craftswomen could buy things for themselves they would otherwise have been unable or reluctant to afford. The craftswomen perceived that the increased income could help them provide for their families, solve problems at home, and avoid interpersonal conflicts. Some pilot work in rural Cote d'Ivoire [37] has suggested that the ability for women to own assets, such as land, meant that they were more independent and had a higher chance to contribute to the household. This financial empowerment was associated with family harmony.

Psychological empowerment was also perceived as an important result of corrected near vision. Interestingly, craftswomen placed importance on the ability to read to avoid embarrassment. The ability to recite the Qu'ran is linked to a woman's dignity and worth in the Muslim culture. When this ability is compromised, the craftswomen feel they might look illiterate or unable to fulfil their religious responsibilities. Therefore, near-vision spectacle correction would have a direct positive psychological impact on these women. Another element of psychological empowerment resulting from better near vision is that it contributed to their independence and self-confidence, ultimately leading to an improved sense of self-determination in decision-making. The ability to make autonomous decisions is significant because it is the core principle of basic human rights of equality and integrity. Studies have suggested that social-cultural factors have led to women in Zanzibar feeling less empowered [21] and lacking the autonomy to decide how to spend household income.

Zanzibari women holding political and leadership positions are increasing, and efforts such as *Wanawake* Wanaweza [38] have been trying to advance women's leadership and political participation. Some women leaders from Zanzibar are Samia Suluhu Hassan, the President of the Republic of Tanzania and Zawadi Amour Nassor, a Member of the Zanzibar House of

Representatives. These positions had previously only been held by men [39]. Hence, it is encouraging that simple near-vision spectacle correction could be perceived as a catalyst to inspire craftswomen to participate in leadership positions and help them work more efficiently.

The pathway to women's empowerment via corrected near vision could be both straightforward and complex (Fig 1). From the findings, we could observe that the relationship between presbyopic correction and perceived women's empowerment could be straightforward–better near vision could improve work performance. Hence, they are more motivated to engage in more work (personal psychological empowerment). However, at times this relationship could be complex. We hypothesise that the complexity arises from the interactions between factors, such as political, psychological, social and economic, that could lead to the perceptions of different aspects of empowerment. For example, better near vision could help craftswomen read by themselves and not be embarrassed (personal psychological empowerment). As a result, they could participate in leadership roles and be more efficient in their political endeavour (personal political empowerment).

The Zanzibari government is committed to achieving gender equality, gender equity, and women's empowerment. The high burden of uncorrected presbyopia, women's disadvantaged position in society, and the importance of near vision to craftswomen led to the conception of Women's Empowerment through Investing in Zanzibari Craftswomen's Eyesight (WE-ZACE) project [40]. The WE-ZACE project is a 6-month longitudinal study to understand how better near vision impacts the empowerment of women. The findings from the WE-ZACE project will provide supportive evidence for policymakers and other stakeholders to increase the provision of eye care services to women.

## Limitations

First, during the time that interviews were conducted in this study, it was impossible to keep track of new emerging ideas using a diary or field notes due to logistical and time constraints during the COVID-19 pandemic. Therefore, deciding *a priori* on adequate sample size was an alternative method to determine the number of participants for the study. Second, craftswomen did not fulfil the quota in each age group included in the sampling frame due to unavailability, resulting in a larger proportion of craftswomen between the ages of 40 and 55 than those over 55. Therefore, age differences could have influenced some perceptions since older craftswomen could have had presbyopia for longer. Third, only the craftswomen's age and types of craft were analysed and discussed. Other demographics, such as marital status and education level, could have introduced further insights into possible differences in experiences of presbyopia and perceptions of women's empowerment amongst craftswomen.

## Conclusion

This novel study shows that older craftswomen perceived that correcting near vision could empower them at personal and relational levels that encompass economic, psychological, social, political and educational empowerment. Our findings laid the foundation for understanding this under-studied but complex area. Further longitudinal interventional studies are needed to assess how presbyopia correction improves women's empowerment.

## Supporting information

**S1 File. Inclusivity in global research checklist.**
(DOCX)

## Acknowledgments

We thank all the data collectors, craftswomen, and cooperative managers for participating in the study.

## Author Contributions

**Conceptualization:** Fatma Omar, Omar Othman, Ai Chee Yong, Carlos Price-Sanchez, Ronnie Graham, Adrianna Farmer, Eden Mashayo, Ving Fai Chan.

**Data curation:** Michelle Fernandes Martins, Fatma Omar, Omar Othman, Ai Chee Yong, Ving Fai Chan.

**Formal analysis:** Michelle Fernandes Martins, Fatma Omar, Omar Othman, Damaris Mulewa, Christine Graham, Ving Fai Chan.

**Funding acquisition:** Fatma Omar, Omar Othman, Ving Fai Chan.

**Investigation:** Fatma Omar, Omar Othman, Carlos Price-Sanchez, Ronnie Graham, Adrianna Farmer, Eden Mashayo, Ving Fai Chan.

**Methodology:** Fatma Omar, Omar Othman, Gianni Virgili, Ai Chee Yong, Carlos Price-Sanchez, Ronnie Graham, Adrianna Farmer, Eden Mashayo, Ving Fai Chan.

**Project administration:** Fatma Omar, Omar Othman, Eden Mashayo, Ving Fai Chan.

**Supervision:** Fatma Omar, Eden Mashayo.

**Validation:** Christine Graham.

**Visualization:** Michelle Fernandes Martins, Fatma Omar, Omar Othman, Ving Fai Chan.

**Writing – original draft:** Michelle Fernandes Martins, Fatma Omar, Omar Othman, Ving Fai Chan.

**Writing – review & editing:** Michelle Fernandes Martins, Fatma Omar, Omar Othman, Gianni Virgili, Ai Chee Yong, Damaris Mulewa, Christine Graham, Carlos Price-Sanchez, Ronnie Graham, Adrianna Farmer, Eden Mashayo, Ving Fai Chan.

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
