## [Decision Letter · Decision Letter 0]

30 Mar 2023

PONE-D-23-02677How does a pair of near-vision spectacle correction empower older Zanzibari craftswomen?: a qualitative study on perceptionPLOS ONE

Dear Dr. Chan

Thank you for submitting your manuscript to PLOS ONE. After careful consideration, we feel that it has merit but does not fully meet PLOS ONE’s publication criteria as it currently stands. Therefore, we invite you to submit a revised version of the manuscript that addresses the points raised during the review process.

Surya Bahadur Parajuli, MD

Academic Editor

PLOS ONE

“MFM.

This study formed part of MFM's master's dissertation.”

“We want to thank the Novartis (Excellence in Ophthalmology and Vision Award, XOVA) funded the project [grant number NPO 6240 R8898CPH]. The funder has not contributed to the design and the analysis of the study.”

“YES.

VFC.

NPO 6240 R8898CPH

Novartis (Excellence in Ophthalmology and Vision Award, XOVA)

https://www.xova.novartis.com/#dwell

The funder did not play any role in the study design, data collection and analysis, decision to publish, or preparation of the manuscript”

Additional Editor Comments:

Dear Authors

Thank you for your submission. This a good manuscript in its field. I have outlined the comments provided by reviewers.

Reviewer 1:

I have mentioned few suggestions below:

Abstract

1. Keywords: I suggest authors to arrange in alphabetical order.

2. Introduction: In line 60 and 78, kindly check the intext citation format. It should be similar. Also, in text citation should be in serial order.

Reviewer 2

Line 28: Kindly include your study duration.

Line 40: You can add your major finding as per your objective.

Line 46-123: Gross comment for Introduction section: Be more specific as per your objective. You can rearrange many points included in the introduction section to methodology section. It seems better. Writing all the methodological details will misguide the readers and they might be not interested. So, be specific in your introduction. Better to use 125 words.

Line 125: add study duration, you need to start with type of research design, approach etc not from ethical approval/ You need restructuring of your method section.

Line 195: Result section has few typo errors, kindly correct. Somewhere the syntax is not maintained. Kindly improve your syntax.

In your discussion, you need to add other social, political and economic factors that might be confounding variables. You need cautious interpretation.

In your conclusion: better not to write a conclusion which was beyond your scope of research. Actually, that needs further research.

In your reference, some typo errors noted, kindly rectify.

Kindly go through the comments and rectify your manuscript.

Reviewers' comments:

Reviewer's Responses to Questions

**Comments to the Author**

1. Is the manuscript technically sound, and do the data support the conclusions?

Reviewer #1: Yes

Reviewer #2: Yes

2. Has the statistical analysis been performed appropriately and rigorously? 

Reviewer #1: Yes

Reviewer #2: Yes

3. Have the authors made all data underlying the findings in their manuscript fully available?

Reviewer #1: Yes

Reviewer #2: Yes

4. Is the manuscript presented in an intelligible fashion and written in standard English?

Reviewer #1: Yes

Reviewer #2: Yes

5. Review Comments to the Author

Reviewer #1: I have mentioned few suggestions below:

Abstract

1. Keywords: I suggest authors to arrange in alphabetical order.

2. Introduction: In line 60 and 78, kindly check the intext citation format. It should be similar. Also, in text citation should be in serial order.

Reviewer #2: Line 28: Kindly include your study duration.

Line 40: You can add your major finding as per your objective.

Line 46-123: Gross comment for Introduction section: Be more specific as per your objective. You can rearrange many points included in the introduction section to methodology section. It seems better. Writing all the methodological details will misguide the readers and they might be not interested. So, be specific in your introduction. Better to use 125 words.

Line 125: add study duration, you need to start with type of research design, approach etc not from ethical approval/ You need restructuring of your method section.

Line 195: Result section has few typo errors, kindly correct. Somewhere the syntax is not maintained. Kindly improve your syntax.

In your discussion, you need to add other social, political and economic factors that might be confounding variables. You need cautious interpretation.

In your conclusion: better not to write a conclusion which was beyond your scope of research. Actually, that needs further research.

In your reference, some typo errors noted, kindly rectify.

6. PLOS authors have the option to publish the peer review history of their article (what does this mean?). If published, this will include your full peer review and any attached files.

Reviewer #1: No

Reviewer #2: **Yes: **Heera KC

---

## [Author Response · Author response to Decision Letter 0]

6 May 2023

1. Please ensure that your manuscript meets PLOS ONE’s style requirements, including those for file naming. The PLOS ONE style templates can be found at

Response: Noted with changes made.

Response: We have now uploaded the completed form as Supporting Information. 

“MFM.

This study formed part of MFM’s master’s dissertation.”

Please confirm that this does not alter your adherence to all PLOS ONE policies on sharing data and materials, by including the following statement: “This does not alter our adherence to PLOS ONE policies on sharing data and materials.” (as detailed online in our guide for authors http://journals.plos.org/plosone/s/competing-interests). If there are restrictions on sharing of data and/or materials, please state these. Please note that we cannot proceed with consideration of your article until this information has been declared.

Response: We have included the additional clause in the manuscript and cover letter.

“We want to thank the Novartis (Excellence in Ophthalmology and Vision Award, XOVA) funded the project [grant number NPO 6240 R8898CPH]. The funder has not contributed to the design and the analysis of the study.”

“YES.

VFC.

NPO 6240 R8898CPH

Novartis (Excellence in Ophthalmology and Vision Award, XOVA)

https://www.xova.novartis.com/#dwell

The funder did not play any role in the study design, data collection and analysis, decision to publish, or preparation of the manuscript”

Response: Amended.

Response: We have now provided the doi number and the link to the dataset. (https://doi.org/10.17034/d32d197f-87fa-4c4c-bc08-b80e2e1c366d)

Response: We have checked the reference list to ensure its completeness and accuracy. 

Additional Editor Comments:

Dear Authors

Thank you for your submission. This a good manuscript in its field. I have outlined the comments provided by reviewers.

Reviewer 1:

I have mentioned few suggestions below:

Abstract

1. Keywords: I suggest authors to arrange in alphabetical order.

Response: Amended. 

2. Introduction: In line 60 and 78, kindly check the intext citation format. It should be similar. Also, in text citation should be in serial order.

Response: Amended. We have standardise the citation format and made sure they are in serial order.

Reviewer 2

Line 28: Kindly include your study duration.

Response: Yes. Included study duration in Line 28.

Line 40: You can add your major finding as per your objective.

Response: Amended to include the major finding (Older craftswomen perceived that correcting near vision could empower them at personal and relational levels that encompass economic, psychological, social, political and educational empowerment). 

Line 46-123: Gross comment for Introduction section: Be more specific as per your objective. You can rearrange many points included in the introduction section to methodology section. It seems better. Writing all the methodological details will misguide the readers and they might be not interested. So, be specific in your introduction. Better to use 125 words.

Response: Thank you for the comment. We streamlined the introduction section by removing information that might be too much for the readers and moving some content to the methods section. We also ensured that the flow of the content is logical and relevant to the subject matter. 

Line 125: add study duration, you need to start with type of research design, approach etc not from ethical approval/ You need restructuring of your method section.

Response: We have included the study duration in Line 165. We have restructured the methods section as best as possible by balancing Reviewer 2’s comments, PLoS ONE’s guidelines, the COREQ checklist and sample PLoS ONE publications on similar qualitative research. 

Line 195: Result section has few typo errors, kindly correct. Somewhere the syntax is not maintained. Kindly improve your syntax.

Response: Thank you for pointing out the typos and syntax errors. They are now amended.

In your discussion, you need to add other social, political and economic factors that might be confounding variables. You need cautious interpretation.

Response: Thank you very much for this excellent point. Indeed, the visual map (previously as supplementary material, now made into Figure 1 as part of the manuscript) shows that the pathway to women’s empowerment via corrected near vision could be both straightforward and complex. The complexity is seen as the interactions between all the factors mentioned by Reviewer 2 that could have confounded and/or led to the perceptions of different aspects of empowerment. We included a paragraph explaining this. 

In your conclusion: better not to write a conclusion which was beyond your scope of research. Actually, that needs further research.

Response: Thank you for the comment. We now streamlined the conclusion to only include the main findings, the implication of the findings and the need for further research.

In your reference, some typo errors noted, kindly rectify.

Response: Amended.

---

## [Editor Report · Decision Letter 1]

15 May 2023

How does a pair of near-vision spectacle correction empower older Zanzibari craftswomen?: a qualitative study on perception

PONE-D-23-02677R1

Dear Dr. Ving Fai Chan

We’re pleased to inform you that your manuscript has been judged scientifically suitable for publication and will be formally accepted for publication once it meets all outstanding technical requirements.

Kind regards,

Surya Bahadur Parajuli, MD

Academic Editor

PLOS ONE

---

## [Editor Report · Acceptance letter]

17 May 2023

PONE-D-23-02677R1 

How does a pair of near-vision spectacle correction empower older Zanzibari craftswomen?: a qualitative study on perception 

Dear Dr. Chan:

I'm pleased to inform you that your manuscript has been deemed suitable for publication in PLOS ONE. Congratulations! Your manuscript is now with our production department. 

Kind regards, 

on behalf of

Dr. Surya Bahadur Parajuli 

Academic Editor

PLOS ONE